# Single Cell Killing Kinetics Differentiate Phenotypic Bacterial Responses to Different Antibacterial Classes

Yuewen Zhang,[a,b] Ibolya Kepiro,[b] Maxim G. Ryadnov,[b,c] Stefano Pagliara[a]

[a]Living Systems Institute and Biosciences, University of Exeter, Exeter, United Kingdom
[b]National Physical Laboratory, Teddington, United Kingdom
[c]Department of Physics, King's College London, London, United Kingdom

**ABSTRACT** With the spread of multidrug-resistant bacteria, there has been an increasing focus on molecular classes that have not yet yielded an antibiotic. A key capability for assessing and prescribing new antibacterial treatments is to compare the effects antibacterial agents have on bacterial growth at a phenotypic, single-cell level. Here, we combined time-lapse microscopy with microfluidics to investigate the concentration-dependent killing kinetics of stationary-phase *Escherichia coli* cells. We used antibacterial agents from three different molecular classes, $\beta$-lactams and fluoroquinolones, with the known antibiotics ampicillin and ciprofloxacin, respectively, and a new experimental class, protein Ψ-capsids. We found that bacterial cells elongated when treated with ampicillin and ciprofloxacin used at their minimum inhibitory concentration (MIC). This was in contrast to Ψ-capsids, which arrested bacterial elongation within the first two hours of treatment. At concentrations exceeding the MIC, all the antibacterial agents tested arrested bacterial growth within the first 2 h of treatment. Further, our single-cell experiments revealed differences in the modes of action of three different agents. At the MIC, ampicillin and ciprofloxacin caused the lysis of bacterial cells, whereas at higher concentrations, the mode of action shifted toward membrane disruption. The Ψ-capsids killed cells by disrupting their membranes at all concentrations tested. Finally, at increasing concentrations, ampicillin and Ψ-capsids reduced the fraction of the population that survived treatment in a viable but nonculturable state, whereas ciprofloxacin increased this fraction. This study introduces an effective capability to differentiate the killing kinetics of antibacterial agents from different molecular classes and offers a high content analysis of antibacterial mechanisms at the single-cell level.

**IMPORTANCE** Antibiotics act against bacterial pathogens by inhibiting their growth or killing them directly. Different modes of action determine different antibacterial responses, whereas phenotypic differences in bacteria can challenge the efficacy of antibiotics. Therefore, it is important to be able to differentiate the concentration-dependent killing kinetics of antibacterial agents at a single-cell level, in particular for molecular classes which have not yielded an antibiotic before. Here, we measured single-cell responses using microfluidics-enabled imaging, revealing that a novel class of antibacterial agents, protein Ψ-capsids, arrests bacterial elongation at the onset of treatment, whereas elongation continues for cells treated with $\beta$-lactam and fluoroquinolone antibiotics. The study advances our current understanding of antibacterial function and offers an effective strategy for the comparative design of new antibacterial therapies, as well as clinical antibiotic susceptibility testing.

**KEYWORDS** antibiotics, *Escherichia coli*, microfluidics, antibiotic susceptibility testing, bacterial membranes, minimum inhibitory concentration, single-cell analysis, ampicillin, antimicrobial resistance, ciprofloxacin, persisters, viable but nonculturable

Address correspondence to Stefano Pagliara, s.pagliara@exeter.ac.uk.

The authors declare no conflict of interest.

Bacteria are single-cell organisms that have inhabited our planet for billions of years and make up 13% of the Earth's biomass (1). They mediate environmental processes that permit life on Earth; their diversity is paramount for bioremediation and biotechnology

(2), and the trillions of bacteria inside us perform functions that are beneficial for us (3). However, some bacteria constitute a threat to humans and animals, causing diseases such as food poisoning, nosocomial infections, pneumonia, and sepsis and a yearly death toll of 5 million people (4). This scenario is predicted to worsen due to bacterial infections that are resistant to antibiotic treatment, with a forecast of an annual gross domestic product shortfall of up to $3.4 trillion by 2030 (5). Therefore, there is an urgent need to improve our understanding of antibacterial mechanisms and their efficacy in clearing bacterial infections.

Antibiotics tackle pathogens by either preventing their growth (i.e., bacteriostatic antibiotics) or directly killing them (i.e., bactericidal antibiotics) (6). The susceptibility of infecting bacteria to antibiotics is generally determined by measuring bacterial growth in the presence of an antibiotic, either in liquid cultures or on agar plates (7). In such assays, the minimum inhibitory concentration (MIC) is measured for a susceptible control strain, and if a strain grows at higher drug concentrations, the strain is considered resistant (8, 9). Since it takes between 24 and 48 h for these phenotypic antibiotic susceptibility tests (ASTs) to provide reliable data, these methods are not suitable for guiding antibiotic treatment in the early stages of an infection. Moreover, existing evidence suggests that conventional MIC assays fail to differentiate antibiotic effects on tolerant populations (10–12) or subpopulations (such as persister cells and viable but nonculturable [VBNC] cells) (13–20) in both laboratory and clinical strains. This evidence indicates that drug susceptibility is much more complex (21) and is driven by phenotypic subpopulations that survive antibiotic treatments and vary in their responses to antibiotics used at different concentrations (14).

Therefore, new phenotypic ASTs have recently been developed to reduce detection times down to 1 to 3 h (22–30). Specifically, time-lapse microscopy coupled with microfluidic devices has been extensively used to investigate the accumulation (31–33), efficacy (18, 34–38), and comparative performance (39–41) of conventional and experimental antibiotics. Notably, Baltekin et al. characterized the susceptibility of urinary tract infection *Escherichia coli* cells to nine different antibiotics within 30 min of loading urine samples in a microfluidic device, while delivering each antibiotic at their breakpoint concentration value (35). However, the nature and extent of variations in the time-kill kinetics of antibiotics used at different concentrations at the single-cell level remain to be investigated.

Here, we combined time-lapse microscopy with a microfluidic device equipped with small parallel channels, each hosting between one and six cells, dubbed a "mother machine" (42), to investigate the impact of increasing concentrations of conventional and experimental antibacterial agents on the killing kinetics of individual stationary-phase *E. coli* cells. We chose to investigate stationary-phase *E. coli* cells because the fraction of phenotypic subpopulations that survive antibiotic treatments in the growth phase is larger than that measured for exponential-phase *E. coli* cells (15, 37). We cross-compared two conventional antibiotics used in clinics, namely, ampicillin and ciprofloxacin from the $\beta$-lactam and fluoroquinolone classes, respectively, and one recently introduced experimental antibacterial agent, termed peptide $\Psi$-capsids. $\Psi$-capsids are viruslike mimetics assembled from a single triskel peptide which favors attack on bacterial membranes (43). This study revealed four different single-cell responses to treatments with these agents: lysed cells, cells with disrupted membranes, cells that maintained membrane integrity but did not grow during or after the treatments (i.e., viable but nonculturable cells) (16, 18, 44), and cells that grew after the treatments (i.e., persisters) (13–15, 17, 19, 20). We found that the rate of elongation and relative abundances of these phenotypes were significantly affected by the concentrations used and the duration of the treatments for ciprofloxacin and ampicillin. In contrast, protein $\Psi$-capsids were found to arrest cell elongation at all concentrations tested.

## RESULTS

**Single-cell responses to inhibitory concentrations of different classes of antibiotics.** We adapted our recent single-cell platform (18, 31, 39, 45, 46) to investigate the response of hundreds of individual stationary-phase *E. coli* cells to ampicillin, ciprofloxacin, and protein $\Psi$-capsids at their MIC (5, 0.125, and 15 $\mu$g/mL, respectively). We found notable differences in response to these three agents within a clonal *E. coli* population: individual *E. coli* cells

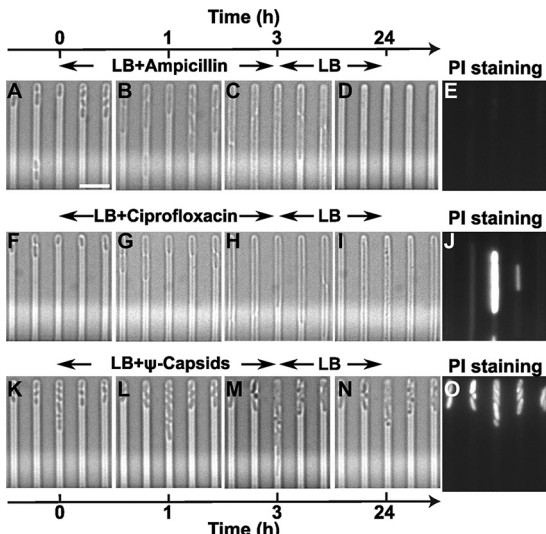

**FIG 1** Single-cell response to antibiotic treatment. Representative bright-field microscopy images of individual *E. coli* cells confined in independent channels of the microfluidic mother machine and responding to ampicillin (A to D), ciprofloxacin (F to I), or Ψ-capsids (K to N), with each drug at its MIC. Antibacterial treatments in the mother machine were carried out from $t = 0$ to $t = 3$ h, with each drug dissolved in lysogeny broth (LB). At $t = 3$ h, each agent was removed from the microfluidic environment, and *E. coli* cells were incubated in LB for another 21 h. A minimum of 500 individual cells collected in biological triplicate were investigated for each treatment. Corresponding fluorescence microscopy images of individual cells that were stained with propidium iodide (PI) at $t = 24$ h after a 3-h treatment with ampicillin (E), ciprofloxacin (J), and Ψ-capsids (O). Bar = 5 $\mu$m.

elongated during a 3-h treatment with ampicillin and ciprofloxacin (Fig. 1A to C and Fig. 1F to H), resembling filamenting phenotypes in the presence of antibiotics from the same classes (20, 47). Bacterial cells were simultaneously supplied with the antibiotic in use and lysogeny broth (LB) medium that permitted bacterial elongation and exit from stationary phase. In contrast, Ψ-capsids arrested cell elongation from the onset of the treatment, with drastic changes to the cell morphology (Fig. 1K to M).

Specifically, when ampicillin was used at the MIC, the majority of *E. coli* cells stopped elongating at $t = 4$ h (i.e., after a 3-h treatment with ampicillin and 1 h of incubation in lysogeny broth [LB]). By the end of the experiment (i.e., 3 h of ampicillin treatment, followed by 21 h of incubation in LB), some cells lysed (Fig. 1D and E; Fig. 2A), while others displayed a compromised membrane when stained with propidium iodide (PI) (Fig. 2B). Noteworthy, *E. coli* cells that had lysed by the end of the experiment reached a significantly higher maximal average elongation rate than cells that displayed a compromised membrane (8.5 ± 6.0 and 3.3 ± 3.6 $\mu$m h$^{-1}$, respectively [$P < 0.05$, according to an unpaired, two-tailed Welch's $t$ test]) at $t = 3$ h, i.e., at the time point when ampicillin was removed from the microfluidic device.

Other cells retained an integral membrane but did not divide within 21 h of incubation in LB medium post-antibiotic exposure (Fig. 2C). These cells resembled the viable but non-culturable (VBNC) phenotype: cells that do not divide in the presence of antibiotics (or other stressors (44, 48, 49)) or after their removal (16, 18, 36, 37, 47, 50, 51). In our experiments, cells resembling the VBNC phenotype did not elongate during the treatment with antibiotics or within the 21 h of incubation in LB medium post-antibiotic exposure (Fig. 2C). However, we cannot exclude that these cells were persisters with a very long lag time period (i.e., longer than 21 h, our period of observation). In fact, lag times greater than 21 h were recently reported for a minority of *E. coli* cells after starvation (52).

Further, it should be noted that since we analyzed only hundreds of individual cells per experiment, we found persister cells, i.e., cells that are able to survive lethal doses of antibiotics and to resume division after antibiotic removal (17, 19, 53–56), only in a minority of the experimental conditions investigated. Therefore, for clarity, we excluded persister cells from the present comparisons of single-cell elongation rates during antibacterial treatments.

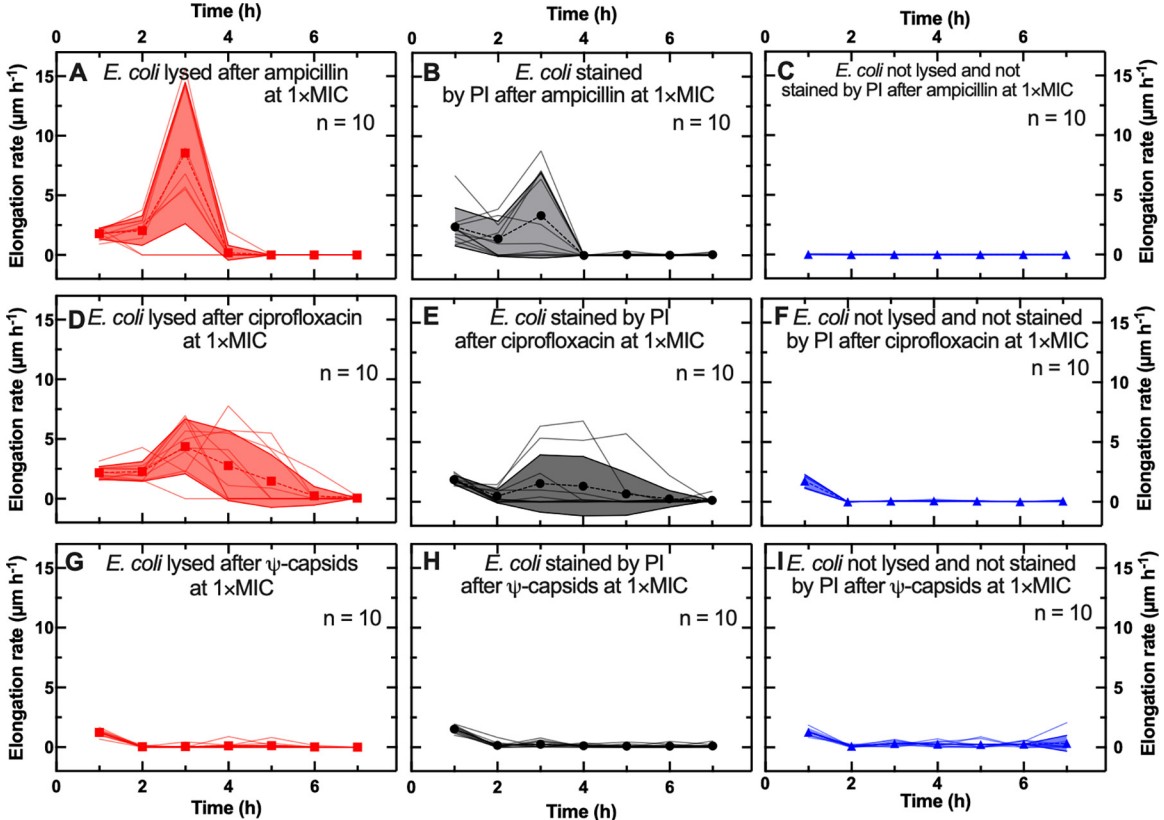

**FIG 2** Single-cell elongation rates during antibacterial treatments at the MIC. Time dependence of single-cell elongation rates for cells that lysed (red solid lines), cells with disrupted membranes (black solid lines), and cells with intact membranes that did not divide (blue solid lines) when treated with ampicillin (A to C), ciprofloxacin (D to F), and Ψ-capsids (G to I) at their respective MICs. The symbols and shaded areas represent the means and standard deviations, respectively, of the single-cell elongation rate values. A total of 10 single-cell temporal dependences of the elongation rates are presented for each phenotype and antibacterial treatment. These trajectories are representative of over 500 individual cells collected in biological triplicate for each experimental condition.

When using ciprofloxacin at the MIC, we recorded similar single-cell responses (Fig. 1F to J) to those measured for ampicillin. However, the *E. coli* cells that were lysed by ciprofloxacin at the end of the experiment reached a significantly lower maximal average elongation rate at $t = 3$ h than that of cells treated with ampicillin ($4.4 \pm 2.3$ and $8.5 \pm 6.0$ $\mu$m h$^{-1}$, respectively [$P < 0.01$]) (Fig. 2D). Cells treated with ciprofloxacin that were lysed by the end of the experiment elongated faster than cells with compromised membranes (with maximal average elongation rates at $t = 3$ h of $4.4 \pm 2.3$ and $1.5 \pm 2.4$ $\mu$m h$^{-1}$, respectively; Fig. 2D and E). In contrast, cells which had intact membranes but did not divide reached a higher average elongation rate at $t = 1$ h (Fig. 2F) than that recorded during ampicillin treatments ($1.7 \pm 0.6$ and $0.02 \pm 0.01$ $\mu$m h$^{-1}$, respectively [$P < 0.01$]).

We recorded very different single-cell responses to Ψ-capsids used at the MIC (Fig. 1L to O). Most cells treated with Ψ-capsids displayed lower average elongation rates at $t = 1$ h than those recorded during the ampicillin and ciprofloxacin treatments: for cells that lysed, $1.2 \pm 0.3$ $\mu$m h$^{-1}$ for the Ψ-capsid-treated cells versus $1.8 \pm 0.4$ $\mu$m h$^{-1}$ ($P < 0.01$) and $2.2 \pm 0.5$ $\mu$m h$^{-1}$ ($P < 0.001$) for cells treated with ampicillin and ciprofloxacin, respectively; for cells with compromised membranes, $1.3 \pm 0.3$ $\mu$m h$^{-1}$ versus $2.4 \pm 1.6$ $\mu$m h$^{-1}$ (not significant [NS]) and $1.9 \pm 0.4$ $\mu$m h$^{-1}$ ($P < 0.05$), respectively; and for nondividing cells with intact membranes, $1.3 \pm 0.3$ $\mu$m h$^{-1}$ versus $0.02 \pm 0.0.01$ $\mu$m h$^{-1}$ ($P < 0.0001$) and $1.7 \pm 0.6$ (NS), respectively. Strikingly, all cells treated with Ψ-capsids stopped growing from $t = 2$ h onwards (Fig. 2G to I), in contrast to the recorded increases in elongation rate during the ampicillin and ciprofloxacin treatments.

**Single-cell responses to supra-MICs.** To gain better insight into the killing kinetics of each agent, the observed differences were then tested as a function of concentration.

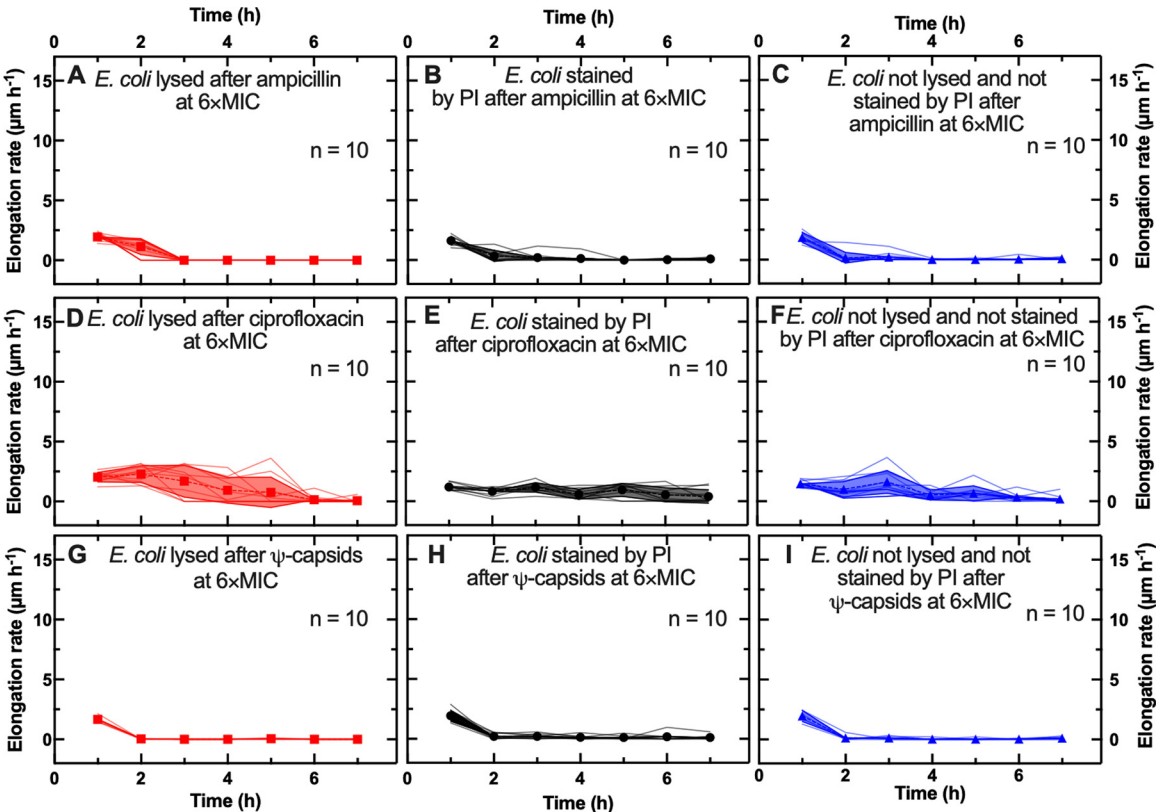

**FIG 3** Single-cell elongation during antibacterial treatments at 6× MIC. Temporal dependence of single-cell elongation rates for cells that lysed (red solid lines), cells that displayed a compromised membrane (black solid lines), and cells with intact membranes that did not divide (blue solid lines) when treated with ampicillin (A to C), ciprofloxacin (D to F), and Ψ-capsids (G to I) at 6× MIC. Symbols and shaded areas represent the means and standard deviations, respectively, of the single-cell elongation rate values. A total of 10 single-cell time dependences of the elongation rates are presented for each phenotype and antibacterial treatment. These trajectories are representative of over 500 individual cells collected in biological triplicate for each experimental condition.

Cells that were treated with ampicillin at 6× MIC and lysed by the end of the experiment showed at $t = 1$ h an elongation rate comparable to that recorded during the ampicillin treatment at the MIC (1.9 ± 0.3 and 1.8 ± 0.4 $\mu$m h$^{-1}$, respectively [NS]; Fig. 3A). Similarly, cells with a compromised membrane showed an elongation rate comparable to that measured during ampicillin treatment at the MIC (1.6 ± 0.4 and 2.4 ± 1.6 $\mu$m h$^{-1}$, respectively [NS]; Fig. 3B). In contrast, cells with intact membranes showed a higher elongation rate than that for ampicillin treatment at the MIC (1.9 ± 0.4 and 0.02 ± 0.01 $\mu$m h$^{-1}$, respectively [$P < 0.0001$]; Fig. 3C). However, whereas during ampicillin treatment at the MIC, both cells that lysed and cells that displayed a compromised membrane increased their elongation rate at $t = 2$ h and $t = 3$ h compared to that at $t = 1$ h (Fig. 2A and B), during ampicillin treatment at 6× MIC, both phenotypes had a significantly reduced elongation rate from $t = 2$ h onwards (from 1.9 ± 0.3 to 1.1 ± 0.6 $\mu$m h$^{-1}$ [$P < 0.01$]; from 1.6 ± 0.4 to 0.4 ± 0.5 $\mu$m h$^{-1}$ [$P < 0.0001$]) and largely stopped elongating at $t = 3$ h. In addition, cells with intact membranes showed a similar temporal dependence of elongation rate (Fig. 3C).

In the case of ciprofloxacin treatments at 6× MIC, cells that lysed by the end of the experiment displayed an elongation rate comparable with that recorded during ciprofloxacin treatment at the MIC at $t = 1$ h (2.0 ± 0.4 versus 2.2 ± 0.5 $\mu$m h$^{-1}$ [NS]; Fig. 3D). In contrast, cells that displayed a compromised membrane by the end of the experiment showed an elongation rate significantly lower than that measured during ciprofloxacin treatment at the MIC (1.9 ± 0.4 and 2.4 ± 1.6 $\mu$m h$^{-1}$, respectively [$P < 0.001$]; Fig. 3E). Similar to the trend observed during ciprofloxacin treatments at the MIC, both bacterial phenotypes continued elongating, albeit more slowly, up to $t = 5$ h. Cells with intact membranes that did not divide at the end of the experiment slowly elongated up to $t = 5$ h (Fig. 3F).

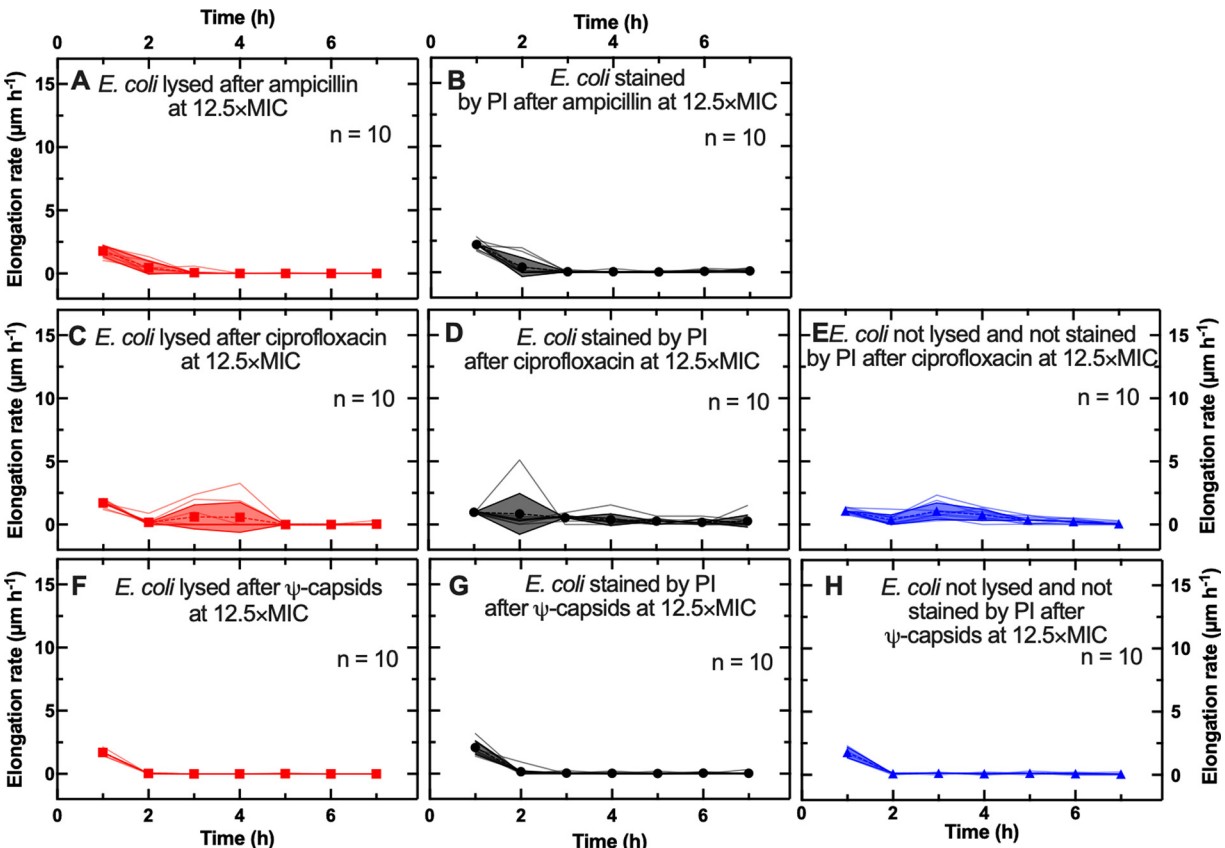

**FIG 4** Single-cell elongation during antibiotic treatment at 12× MIC. Time dependence of single-cell elongation rates for cells that lysed (red solid lines), cells that displayed a compromised membrane (black solid lines), and cells with intact membranes that did not divide (blue solid lines) when treated with ampicillin (A to B), ciprofloxacin (C to E), and Ψ-capsids (F to H) at 12.5× MICs. Symbols and shaded areas represent the means and standard deviations, respectively, of the single-cell elongation rate values. A total of 10 single-cell temporal dependences of the elongation rate are presented for each phenotype and each antibacterial treatment. These trajectories are representative of over 500 individual cells collected in biological triplicate for each experimental condition.

Finally, all cells treated with Ψ-capsids at 6× MIC displayed a time dependence of elongation rates comparable to that recorded for the treatments at the MIC (Fig. 3G to I), suggesting that killing by Ψ-capsids is concentration independent.

Further increasing the antibiotic concentration up to 12.5× MIC did not significantly alter the pattern of elongation rates reported above. Cells that either lysed or displayed a compromised membrane after ampicillin treatments at 12.5× MIC showed maximal elongation rate values at $t = 1$ h comparable to (or higher than) those recorded during ampicillin treatment at 6× MIC (1.8 ± 0.5 versus 2.0 ± 0.4 $\mu$m h$^{-1}$ [NS]; Fig. 4A; 2.2 ± 0.4 versus 1.6 ± 0.4 $\mu$m h$^{-1}$ [$P < 0.01$]; Fig. 4B). Cells that either lysed or displayed a compromised membrane or retained intact membranes after ciprofloxacin treatments at 12.5× MIC showed maximal elongation rate values at $t = 1$ h comparable or lower to those recorded during ciprofloxacin treatments at 6× MIC (1.7 ± 0.3 versus 2.0 ± 0.4 [not significant]; Fig. 4C; 1.0 ± 0.1 versus 1.2 ± 0.3 [$P < 0.05$]; Fig. 4D; 1.5 ± 0.3 versus 1.1 ± 0.2 [$P < 0.01$]; Fig. 4E). All cells displayed comparable elongation rates during Ψ-capsid treatment at 6× and 12.5× MIC (Fig. 3G to I and Fig. 4F to H, respectively). All cells largely arrested elongation from $t = 2$ h onwards after treatment by any of the three agents used (Fig. 4). These data confirm that the Ψ-capsid killing dynamics of *E. coli* is concentration independent, whereas treatments with ampicillin or ciprofloxacin are accompanied by extensive cell elongation at the MIC and elongation arrest increasing at higher concentrations.

**Changes in phenotypic composition at increasing concentrations of antibacterial agents.** With the concentration dependencies of the elongation rate established, we investigated the impact that increasing concentrations of each agent may have on the phenotypic composition of clonal *E. coli* populations. When using ampicillin at the MIC, the fraction

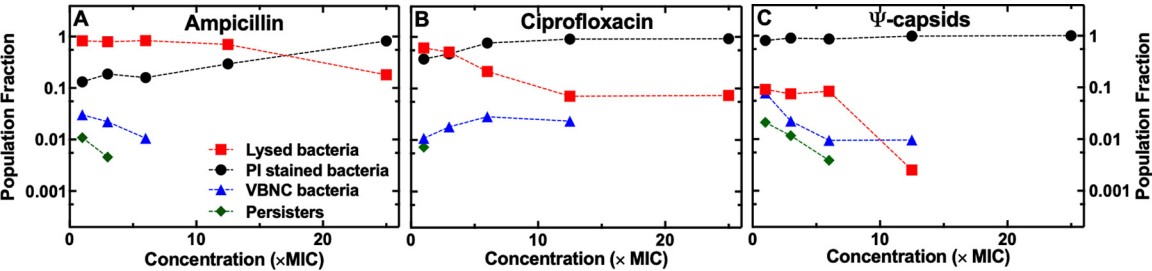

**FIG 5** Concentration-dependent bacterial killing. Dependence of the fractions of cells that lysed (red squares), cells with compromised membranes (black circles), VBNC cells (blue triangles), or persister cells (green diamonds) on the concentration of ampicillin (A), ciprofloxacin (B), and Ψ-capsids (C). The symbols represent the average of fractions measured from over 300 microfluidic channels from 3 different biological replicates and containing over 500 individual cells for each antibacterial treatment tested. The error bars reporting the standard error of the mean associated to each average value are hidden behind the data points due to the large sample size.

of the *E. coli* population that lysed was 0.83 (red squares in Fig. 5A), whereas the fractions of the population that displayed a compromised membrane, or a VBNC or persister phenotype, were 0.13, 0.03, and 0.01, respectively (black circles, blue triangles, and green diamonds in Fig. 5A, respectively). As the concentrations of ampicillin increased, the fraction of cells with compromised membranes increased significantly (Pearson correlation coefficient, 0.96 [$P < 0.01$]; black circles in Fig. 5A), whereas the fraction of cells that lysed significantly decreased ($r = -0.95$ [$P < 0.05$]; red squares in Fig. 5A). In fact, at $25\times$ MIC of ampicillin, the fraction of the *E. coli* population that displayed a compromised membrane was 0.82, whereas the fraction of the *E. coli* population that lysed was only 0.18 (Fig. 5A)—an opposite scenario to that recorded for ampicillin at the MIC. Moreover, the fractions of VBNC and persister cells decreased at increasing ampicillin concentrations, reaching zero at $12.5\times$ MIC and $6\times$ MIC, respectively (blue triangles and green diamonds in Fig. 5A, respectively). These data suggest that the concentration of ampicillin strongly impacts the phenotypic composition of a clonal *E. coli* population, affecting the relative abundance of individual cells of specific phenotypes.

When we used ciprofloxacin at the MIC, we found a more equal split between the fraction of cells that lysed (0.61; red squares in Fig. 5B) and the fraction of cells with compromised membranes (0.37; black circles in Fig. 5B), whereas a lower fraction of cells were VBNC or persister cells (0.01 and 0.01; blue triangles and green diamonds in Fig. 5B, respectively). However, increases in the antibiotic concentrations increased the fraction of cells with compromised membranes, compared with the fraction of cells that lysed (Fig. 5B), which was similar to the trend recorded at increasing ampicillin concentrations. At $25\times$ MIC of ciprofloxacin, the fraction of *E. coli* cells with compromised membranes was 0.93, whereas the fraction of cells that lysed was only 0.07 (Fig. 5B). In contrast to the ampicillin-treated cells, the fraction of VBNC cells significantly increased to 0.03 when ciprofloxacin was used at $6\times$ MIC (blue triangles in Fig. 5B [$P < 0.01$]), whereas no fraction of persister cells was observed above $3\times$ MIC (green diamonds in Fig. 5B).

Nearly no dependence in the relative phenotypic abundance was observed for Ψ-capsids. Cells with compromised membranes constituted the majority of the clonal *E. coli* population already at the MIC of Ψ-capsids (0.81; black circles in Fig. 5C). The fraction of these cells increased up to 1 at $25\times$ MIC. In contrast, the fractions of cells that lysed or were VBNC or persister cells decreased with the concentration of Ψ-capsids from 0.09, 0.08, and 0.02, respectively, at the MIC down to 0 at $25\times$ MIC (red squares, blue triangles, and green diamonds in Fig. 5C, respectively). These data suggest that in contrast to ampicillin or ciprofloxacin, Ψ-capsids do not differentiate between different phenotypes, killing cells by compromising their membranes.

**Weak interdependence between killing kinetics and treatment duration or temperature.** Finally, we sought to investigate the impact of other experimental conditions, namely, treatment duration and temperature, on the relative abundance of the three observed phenotypes. When using ampicillin at the MIC and at 37 °C, we found that increasing the treatment duration slightly increased the fraction of cells that lysed (from 0.62 to 0.83 [NS] when the treatment duration was increased from 3 to 24 h; filled squares in Fig. 6A), slightly decreased

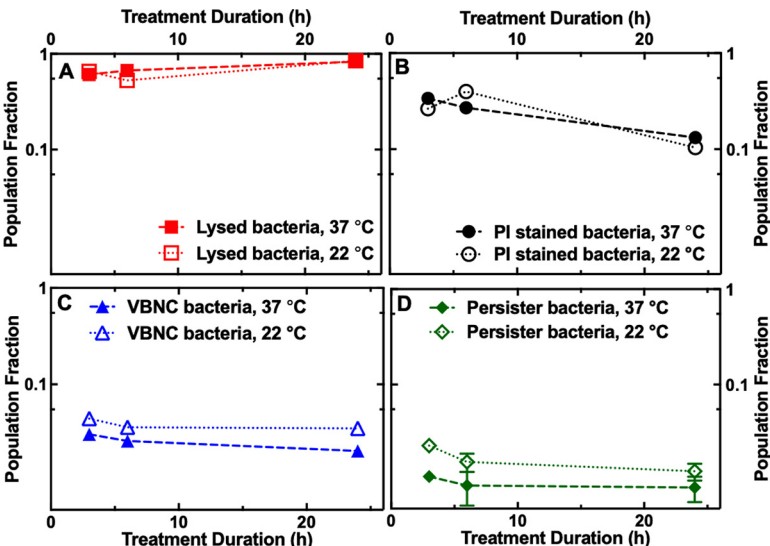

**FIG 6** Duration- and temperature-dependent bacterial killing. Dependence of the fraction of cells that lysed (red squares) (A), cells with compromised membranes (black circles) (B), VBNC calls (blue triangles) (C), and persister cells (green diamonds) (D) on the duration of ampicillin treatments at 22°C or 37°C (open and filled symbols, respectively). The symbols represent the average of fractions measured on over 300 microfluidic channels from 3 different biological replicates and containing over 500 individual cells for each antibacterial treatment. The error bars reporting the standard error of the mean associated to each average value are hidden behind the data points due to the large sample size.

the fraction of cells with compromised membranes (from 0.34 to 0.13 [NS]; filled circles in Fig. 6B), and slightly decreased the fraction of VBNC cells (from 0.05 to 0.03 [NS]; filled triangles in Fig. 6C) but did not have an appreciable impact on the fraction of persister cells (0.01 and 0.01; filled diamonds in Fig. 6D). Therefore, the number of lysed cells significantly decreased with increasing ampicillin concentrations and slightly increased with increasing treatment duration at the MIC of the antibiotic.

The dependence on treatment duration of the fractions of the four observed phenotypes was even less pronounced when the measurements were carried out at room temperature (i.e., 22 °C). In this case, the relative abundance of lysed cells first slightly decreased and then slightly increased with treatment duration (NS) (open squares in Fig. 6A). The fraction of cells with compromised membranes first increased and then decreased (NS) (open circles in Fig. 6B). The fraction of VBNC cells and the fraction of persister cells did not significantly change across the different treatment durations tested (open triangles in Fig. 6C and open diamonds in Fig. 6D, respectively).

## DISCUSSION

The ability to determine bacterial responses to antibacterial agents at the population and individual cell levels is paramount for the development of effective antibacterial therapies. Previous reports described bacterial responses to different antibiotics and their concentrations at the population level (14, 15, 19), whereas published data on comparative single-cell responses to different antibacterial agents used at increasing concentrations were lacking.

Specifically, the use of bulk time-kill assays provided the important insight that treatment of stationary-phase *E. coli* cells with ampicillin produced a relatively higher fraction of persister cells than treatment with ofloxacin (15). Here, we show that although the fraction of *E. coli* persister cells was comparable after treatment with ampicillin and ciprofloxacin when these drugs were used at the MICs, only persisters to ampicillin were found when these two drugs were used at 3× MIC. We also show that experimental agents representing a new class of antimicrobials, Ψ-capsids, led to a fraction of persisters comparable to that recorded after treatment with ampicillin.

Interestingly, whereas the fractions of persister and VBNC cells decreased with increasing concentrations of ampicillin and Ψ-capsids, treatments with increasing concentrations

of ciprofloxacin led to increases in the fraction of VBNC cells at the expense of persister cells. These findings emphasize the importance of developing assays that permit the investigation of persister and VBNC cells simultaneously (16). Noteworthy, the fractions of persister and VBNC cells varied little with the duration and temperature of treatments, e.g., for ampicillin used at the MIC, which reinforces the generality of these findings. These data also agree with findings from previous bulk measurements performed on *Staphylococcus aureus* cells, suggesting that the fraction of persister cells within a clonal population initially decreases with the concentration of the antibiotic used but does not vary further for concentrations above $5\times$ MIC (14). Similarly, we showed that the fraction of VBNC cells initially declines with the drug concentration (except for ciprofloxacin) and does not further change when antibiotics are used above $6\times$ MIC. Our finding that VBNC cells, which possibly entered the VBNC state during nutrient starvation in the stationary phase (57) or during the successive antibiotic treatment (58), outnumber persister cells within clonal stationary-phase *E. coli* populations is consistent with findings by others (59). Finally, it is also interesting that at the highest antibiotic concentration employed (i.e., $25\times$ MIC), we did not detect any VBNC or persister cells. However, it is worth acknowledging that at high antibiotic concentrations (i.e., $>6\times$ MIC), the fraction of persisters and VBNC cells drastically decreased (14); therefore, we might not have been able to capture a statistically meaningful sample of persister and VBNC cells, since we analyzed only 500 individual cells for each experimental condition.

The focus on single-cell measurements allowed us to demonstrate that the mode of action strongly correlates with the molecular class of the agent used and the concentration of the agent. Specifically, when used at the MIC, ampicillin lysed most of the *E. coli* cells, whereas the $\Psi$-capsids disrupted the cell membranes, and treatments with ciprofloxacin led to an equal split between lysed and membrane-compromised cells. The fraction of cells with compromised membranes tended to increase as the concentrations of ampicillin and ciprofloxacin increased, but the fraction of cells that lysed during treatment decreased. At the highest concentrations of each agent used ($25\times$ MIC), most cells were killed by membrane disruption rather than lysis.

A possible explanation for these findings is that at the MICs, cells transiently elongate and produce antibiotic targets, for example, penicillin-binding proteins in the case of ampicillin (60). The inhibited activity of penicillin-binding proteins and impaired formation of peptide cross-links into the cell wall destabilize the cytoplasmic membrane, driving cell lysis (61, 62). At higher drug concentrations, cells cannot resume growth from the stationary phase and do not produce new antibiotic targets beyond those already present in the cells in the stationary phase.

This conjecture is corroborated by our single-cell data, showing that *E. coli* cells that were lysed by ampicillin or ciprofloxacin used at concentrations higher that their MICs displayed significantly lower elongation rates (before lysis) than *E. coli* cells that lysed at the MIC. Further, cells with compromised membranes at these higher concentrations displayed significantly lower elongation rates (before lysis) than cells that lysed. Thus, these findings indicate that ampicillin and ciprofloxacin lyse actively growing cells, but at higher concentrations, these antibiotics cause death in nongrowing or slowly growing cells by membrane disruption. In marked contrast to these conventional antibiotics, $\Psi$-capsids readily penetrate cell membranes, while favoring attack on bacterial membranes at any concentration, causing the rapid disruption of bacterial membranes and drastic changes in the morphology of bacterial cells (43, 63). Such a mechanism does not differentiate between growing and nongrowing cells, as was observed in this study.

Collectively, these data advance our understanding that nongrowing *E. coli* cells are more tolerant than growing cells to conventional antibiotics used at their MICs (64). However, at higher antibiotic concentrations, membrane disruption became the common mode of killing of all three agents investigated. Indeed, $\beta$-lactams can porate cells (65), as well as diffusing via the OmpF porins they use to cross bacterial membranes (66). At high concentrations, ampicillin would enhance the probability of these two scenarios, while ciprofloxacin can exploit nanoscale defects in bacterial membranes, causing damage and leading to cytoplasmic condensation (67).

As a consequence of this study, we reason that for antibacterial testing, single-cell experiments should complement phenotypic ASTs using optical density measurements, which otherwise can be confounded by the presence of dead but not lysed cells (9).

The single-cell experiments described in this study can also provide additional mechanistic insights into the mode of antibacterial action. As an exemplar, cells that ultimately get lysed were shown to transiently elongate during the initial phase of treatment with ampicillin or ciprofloxacin, which is in accord with previous reports on cell filamentation during antibiotic treatment (20, 47). Our data demonstrate that these cells elongate faster when treated with ampicillin than ciprofloxacin at the MIC and that cells with intact membranes that do not divide transiently grow faster during ciprofloxacin treatment. Remarkably, all cells within a clonal *E. coli* population stopped elongating within a 2-h treatment with Ψ-capsids at all concentrations investigated without evidence of filamentation. These data suggest that the killing of *E. coli* cells by Ψ-capsids is concentration independent, whereas treatments with ampicillin and ciprofloxacin are associated with the elongation and filamentation of bacterial cells at the MIC and increasingly rapid arrest of bacterial elongation to the point of membrane disruption at higher concentrations.

Overall, this study offers an effective strategy to better inform methods of potentiating the lethality of antibacterial agents, known and emerging, and the results obtained demonstrate the efficiency of performing cross-comparisons between known and experimental molecular classes.

## MATERIALS AND METHODS

**Chemicals and cell culture.** All chemicals were purchased from Thermo Fisher Scientific or Sigma-Aldrich unless otherwise stated. Lysogeny broth (LB) medium (10 g/L tryptone, 5 g/L yeast extract, and 10 g/L NaCl) and LB agar plates (LB with 15 g liter$^{-1}$ agar) were used for planktonic growth and for setting up overnight cultures or performing CFU assays, respectively. *E. coli* BW25113 cells were purchased from Dharmacon (GE Healthcare). Overnight cultures were prepared by picking a single colony of *E. coli* BW25113 from a streak plate and growing it in 100 mL fresh LB medium in a shaking incubator at 200 rpm and 37°C for 17 h. Stock solutions of ampicillin were prepared in Milli-Q water. Stock solutions of ciprofloxacin were prepared in 0.1-M HCl in Milli-Q water. Ψ-capsids were prepared at a total peptide concentration of 200 $\mu$M in filtered (0.22 $\mu$m) 10 mM MOPS (morpholinepropanesulfonic acid), pH 7.4, at room temperature, as described elsewhere (39). The preparations of the assembled Ψ-capsids were used as stock solutions to generate working concentrations for the kinetic studies.

**Determination of MICs.** Single colonies of *E. coli* BW25113 were picked and cultured overnight in cation-adjusted Mueller-Hinton broth (CAMHB) at 37°C and then diluted 40-fold and grown to an optical density at 600 nm (OD$_{600}$) of 0.5. An aliquot (60 $\mu$L) of each antibiotic stock was added to the first column of a 96-well plate. CAMHB (40 $\mu$L) was added to the first column and 30 $\mu$L CAMHB to all other wells. An aliquot (70 $\mu$L) of solution was then withdrawn from the first column and serially transferred to the next column until 70 $\mu$L solution withdrawn from the last column was discharged. The mid-log-phase cultures (i.e., OD$_{600}$ = 0.5) were diluted to 10$^6$ CFU mL$^{-1}$, and 30 $\mu$L was added to each well, to give a final concentration of 5 × 10$^5$ CFU mL$^{-1}$. Each plate contained two rows of 12 positive-control experiments (i.e., cells growing in CAMHB without antibiotics) and two rows of 12 negative-control experiments (i.e., CAMHB only). The plates were incubated at 37°C overnight. The MICs of the antibacterial agents used in the study were determined against *E. coli* BW25113, with the MIC being the lowest concentration with no visible growth (compared to the positive-control experiments).

**Fabrication of the microfluidic devices.** The mold for the mother machine microfluidic device (42) was obtained by pouring epoxy onto a polydimethylsiloxane (PDMS; Dow Corning) replica of the original mold (kindly provided by S. Jun). This chip is equipped with hundreds of lateral microfluidic channels with a width and height of around 1 $\mu$m and a length of 25 $\mu$m. These lateral channels are connected to a main microfluidic chamber that is 25 $\mu$m high and 100 $\mu$m wide. PDMS replicas of this device were realized as previously described (68). Briefly, a 9:1 (base/curing agent) PDMS mixture was cast on the mold and cured at 70°C for 120 min in an oven. The cured PDMS was peeled from the epoxy mold, and fluidic accesses were created using a 0.75-mm biopsy specimen punch (Harris Uni-Core, WPI). The PDMS chip was irreversibly sealed on a glass coverslip by exposing both surfaces to oxygen plasma treatment (10 s exposure to 30 W plasma power; plasma etcher; Diener, Royal Oak, MI) as previously described (69). This treatment temporarily rendered the PDMS and glass hydrophilic, so within 5 min of bonding, the chip was filled with 2 $\mu$L of a 50-mg/mL bovine serum albumin solution and incubated at 37°C for 30 min, thus passivating the device's internal surfaces and decreasing subsequent cell adhesion.

**Microfluidics-microscopy assay to measure killing kinetics of individual cells.** An overnight culture was prepared as described above, and a 50-mL aliquot was centrifuged for 5 min at 4,000 rpm and 37°C. The supernatant was filtered twice (medical Millex-GS filter, 0.22 $\mu$m; Millipore Corp.) to remove bacterial debris and used to resuspend the cells in their spent LB to an OD$_{600}$ value of 75. A 2-$\mu$L aliquot of this suspension was injected into the mother machine device and incubated at 37°C. The high bacterial concentration favored cells entering the narrow lateral channels from the main microchamber of the mother machine (18). An average of over 300 lateral channels of the mother machine device were filled with bacteria in each experiment carried out. The microfluidic device was completed by the integration of fluorinated ethylene propylene tubing (1/32 in. by

0.008 in.). The inlet tubing was connected to the inlet reservoir, which was connected to a computerized pressure-based flow control system (MFCS-4C; Fluigent) as previously described (70). This instrumentation was controlled by MAESFLO software (Fluigent). At the end of the 20-min incubation period, the chip was mounted on an inverted microscope (IX73; Olympus, Tokyo, Japan), and the cells remaining in the main microchamber of the mother machine were washed into the outlet tubing and into the waste reservoir by flowing each antibiotic solution at 300 $\mu$L h$^{-1}$ for 8 min and then at 100 $\mu$L h$^{-1}$ for 3 h. The solutions of each antimicrobial agent were prepared by diluting each stock solution at the appropriate concentration in LB. Bright-field images were acquired every hour. Images were collected via a 60$\times$ 1.2 numerical aperture (NA) objective (UPLSAPO60XW; Olympus) and a sCMOS camera (Zyla 4.2; Andor, Belfast, UK). The region of interest of the camera was adjusted to visualize 23 lateral channels per image, and images of 50 different areas of the microfluidic device were acquired at each time point in order to collect data from at least 500 individual cells per experiment. The device was moved by two automated stages (M-545.USC and P-545.3C7, for coarse and fine movements, respectively; Physik Instrumente, Karlsruhe, Germany). After a 3-h antibiotic treatment, the microfluidic environment was changed by flowing LB medium at 300 $\mu$L h$^{-1}$ for 8 min and then at 100 $\mu$L h$^{-1}$ for 21 h, acquiring bright-field images every hour for the four 3-h periods and after 21 h. At this time point of the assay, live/dead staining (59) was performed by flowing SYTO9 and propidium iodide (PI; Thermo Fisher Scientific) into the microfluidic device for 25 min and 10 min, respectively, at concentrations of 3.34 $\mu$M and 30 $\mu$M, respectively, according to the manufacturer specifications. Bright-field and fluorescence images were acquired to determine whether each bacterium was dead or alive. SYTO9 and PI staining were imaged using 60% blue and 100% green LED intensity, FITC (fluorescein isothiocyanate) and TRITC (tetramethyl rhodamine isocyanate) filters, respectively, and an exposure time of 0.01 s. The entire assay was carried out at 37°C in an environmental chamber (Solent Scientific, Portsmouth, UK) housing the measurement equipment unless otherwise specified.

**Image and data analysis.** Images were processed using ImageJ software as described elsewhere (71–73), tracking each individual bacterium throughout the duration of the experiment. Briefly, a rectangle was drawn around each bacterium in each bright-field image at every time point, obtaining its width, length, and relative position in the hosting microfluidic channel. An average elongation rate for each bacterium was calculated as an average of the ratios of the differences in bacterial length over the lapse of time between two consecutive time points. Each bacterium was visually categorized as lysed or as having a compromised membrane (if stained with PI) or as displaying an intact membrane but nongrowing at the end of the experiment. Statistical significance was tested using unpaired, two-tailed Welch's $t$ tests. Pearson correlations, means, standard deviations, coefficients of variation, and medians were calculated using GraphPad Prism 9.

**Data availability.** We have made available a step-by-step experimental protocol for the fabrication and handling of microfluidic devices for investigating the interactions between the antimicrobial agents and individual cells (74). Data supporting the conclusions of this article will be made available by the authors to any qualified researcher upon request.

## ACKNOWLEDGMENTS

We acknowledge funding from the Biotechnology and Biological Sciences Research Council (grant BB/V008021/1 to S.P.), the Medical Research Council (grant MCPC17189 to S.P. and M.G.R.), and the UK's Department for Business, Energy and Industrial Strategy (to M.G.R.). We thank Smita Gunnoo for assistance with peptide synthesis.

S.P. and M.G.R. conceived, designed, and supervised the study. Y.Z. and I.K. performed the study. All authors analyzed and interpreted the data. S.P and M.G.R. wrote the manuscript. All authors contributed to compiling and editing the manuscript.

We have no conflicts of interest to declare.

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
