## [Reviewer comments · Microbiology Spectrum]

Microbiology Spectrum

Single cell killing kinetics differentiate phenotypic bacterial responses to different antibacterial classes

Stefano Pagliara, Yuewen Zhang, Ibolya Kepiro, and Maxim Ryadnov

Corresponding Author(s): Stefano Pagliara, University of Exeter

Review Timeline:

Submission Date:	September 9, 2022
Editorial Decision:	October 23, 2022
Revision Received:	November 22, 2022
Accepted:	December 19, 2022

Editor: Minsu Kim

Reviewer(s): The reviewers have opted to remain anonymous.

Transaction Report:

DOI: <https://doi.org/10.1128/spectrum.03667-22>

October 23, 2022

Dr. Stefano Pagliara
University of Exeter
Living Systems Institute
Stocker road
Exeter EX4 4QD
United Kingdom

Re: Spectrum03667-22 (Single cell killing kinetics differentiate phenotypic bacterial responses to different antibacterial classes)

Dear Dr. Stefano Pagliara:

Thank you for submitting your manuscript to Microbiology Spectrum. Your manuscript was reviewed by two referees. Please see their comments below. While they find that your work would be useful to the community, some important points are raised. I request that you constructively address these concerns in the form of a revised manuscript before we make a final decision on publication.

Link Not Available

Sincerely,

Minsu Kim

Journals Department
Reviewer comments:

Reviewer #1 (Comments for the Author):

In this study, the authors investigated the impact of conventional (ampicillin and ofloxacin) and unconventional (Ψ -capsids) antibiotics on *E. coli* killing kinetics at the single-cell level using time-lapse microscopy and a microfluidic device. Their study revealed four different single-cell phenotypes formed in response to treatments: lysed cells; cells with disrupted membranes; viable but non-culturable cells and persisters. While the rate of elongation and relative abundances of the identified phenotypes were found to be dependent on antibiotic concentrations and treatment durations for ciprofloxacin and ampicillin, these

characteristics were not observed in Ψ -capsids-treated cells which were found to be at the growth-arrested state. The study, which introduced an effective way to differentiate the killing kinetics of antibacterial agents, is well-written and justified. I have only a few minor issues:

- Line 109: 6 different MIC values were listed for 3 different antibiotics. Please clarify this.
- Figures 2, 3, and 4: I am assuming the lines represent the cells. Please indicate the number of cells (n) for each panel, as it is not clear in some cases (e.g., Figure 2, panel c).
- Asterisks (*, **, etc.) have been used throughout the text for the statistical analysis. Please, simply define the test type (e.g., two-tailed, Welch's t-test) and P threshold values ($P < 0.05$, $P < 0.001$, etc.) in the text (instead of using asterisks).
- Results associated with Figure 5: Limited number of cells have been analyzed (~500). Although the data is very useful for quantifying the population fractions of lysed and PI-stained cells, the number of cells may not be enough to provide a statistically meaningful quantification for rare cells such as persisters and VBNC cells. The authors should mention this.
- Citations given in Lines 141-142: The studies of Dr. Wood (cited) focused on cell cultures that have been starved in PBS for weeks. The phenotypes they obtained are not VBNC cells. In fact, they are dead cells with empty cytosols. These cells can't be stained by PI because they don't have nucleotides. Unfortunately, their claims are not well justified. It has been identified many times by many groups that VBNC phenotypes with intact membranes and dense cytosols exist in normal cell cultures and they can be metabolically active. I would simply remove these citations from the text since their experimental conditions are not relevant to this study.
- Higher concentrations of ampicillin and ofloxacin have reduced VBNC cell levels (almost zero, according to Figure 5). This is so interesting, any comment on this?

Reviewer #2 (Comments for the Author):

In their manuscript "Single cell killing kinetics differentiate phenotypic bacterial responses to different antibacterial classes", Zhang and colleagues describe responses to different antibiotics at different concentrations, measuring elongation rates and outcomes at the single-cell level. They show that their microfluidics-based method can sensitively detect impacts of antibiotics on cell elongation, membrane permeability, and lysis. These measurements can suggest insights into mechanisms of action for killing or survival of the antibiotics. This is a descriptive study that is generally well-described and of potential interest to a broad audience of microbiologists interested in antibiotic impacts on bacteria. I have only a few minor concerns.

1. The fact that all experiments were performed on stationary phase cells seems very important for all discussion and conclusions to be drawn from the results. It seems worth mentioning this even in the abstract and introduction and explaining the reasoning behind choosing stationary phase cells for this investigation. Related to this point, the abstract and results state that the capsids arrest growth within 2 hours (lines 24 and 113). Discussion of the ampicillin and ciprofloxacin phenotypes always refer to elongation rather than growth, and it seems that this may be a better word choice in the context of stationary phase cells. It is not clear why growth, driven by new biosynthesis, should be occurring in stationary phase cells following antibiotic treatment, and elongation could alternatively be explained by turgor pressure acting on a weakened cell wall for example. It seems outside the scope of this paper to present evidence for one mechanism or another to explain the elongation phenotype, but it is worth explaining very clearly the context. These are all cells that have entered stationary phase and likely depleted an essential nutrient, and size increases could be explained by ongoing low levels of new biosynthesis, by changes in the cell wall allowing turgor-driven expansion, by changes in osmotic balance, or by some combination of these factors.

2. I find the distinction between VBNC and persister cells somewhat problematic. I know that this is following on from previously published work which used a similar designation. However, even that previous work also showed that "VBNC" and "persister" cells share many characteristics. The designation as "VBNC" or "persister" here is made based on whether a cell with an intact membrane after antibiotic exposure has started growing and dividing again or not at one time point (21 h) after removal of antibiotics. Others have shown that resumption of growth following antibiotic exposure can have a very long lag time (longer than 21 h), and that the distribution of lag times is not bimodal (for example, doi: 10.1038/s41586-021-04114-w). There is no clear reason why the 21 h time point should be chosen as a threshold to distinguish between "persisters" (have resumed growth) and "VBNCs" (have not yet resumed growth). It seems possible that some of the VBNCs could instead be called persisters if the chosen time point for assessing resumption of growth was a bit later. My main point is that lag time before resumption of growth after surviving antibiotic exposure has been shown to have a broad, continuous distribution, but using a single time point to assess resumption of growth and assign cells to one of two categories suggests instead a binary outcome. It is interesting and relevant to evaluate the impact of antibiotic type and concentration on survival and resumption of growth, but I think the use of the "persister" and "VBNC" categories is an oversimplification. To address this, I would suggest one of two options: 1) check whether the proportions of "VBNCs" and "persisters" remain unchanged at longer timepoints after antibiotic removal. If these designations remain stable for 36 or 48 hours after antibiotic removal, then this is outside the lag times previously reported and it is more convincing that the "VBNCs" really will not re-grow. 2) Add text to address the difficulty of designating any cell a "VBNC" with any certainty and acknowledge that any cell that still has an intact membrane but has not divided after 21 hours might be an example of a persister with a very long lag time, consistent with times that have been reported by others.

3. The asterisks used to denote significance do not seem to be explained anywhere (apologies if I have missed this). Which p-values do each number of asterisks refer to? I am not sure what the asterisks mean in the context of r values (lines 258-259 for example).

Staff Comments:

Preparing Revision Guidelines

Please return the manuscript within 60 days; if you cannot complete the modification within this time period, please contact me. If you do not wish to modify the manuscript and prefer to submit it to another journal, please notify me of your decision immediately so that the manuscript may be formally withdrawn from consideration by Microbiology Spectrum.

We would like to thank both reviewers for their positive comments and suggestions on how to improve our manuscript. Please note that we refer below to the document titled “Marked Up Manuscript - For Review Only” where we have underlined our revisions to facilitate review.

Reviewer 1.

1. *Line 109: 6 different MIC values were listed for 3 different antibiotics. Please clarify this.*

In our submitted manuscript we reported the MIC value for each antibiotic both in terms of concentration and molarity. We understand that this presentation might create confusion and we have therefore retained only MIC values in terms of concentration in our revised manuscript.

This issue has been addressed on line 112.

2. *Figures 2, 3, and 4: I am assuming the lines represent the cells. Please indicate the number of cells (n) for each panel, as it is not clear in some cases (e.g., Figure 2, panel c).*

We have now indicated the number of cells n in each panel of Figures 2, 3 and 4.

3. *Asterisks (*, **, etc.) have been used throughout the text for the statistical analysis. Please, simply define the test type (e.g., two-tailed, Welch's t-test) and P threshold values ($P < 0.05$, $P < 0.001$, etc.) in the text (instead of using asterisks).*

We have implemented this suggested change throughout the manuscript.

4. *Results associated with Figure 5: Limited number of cells have been analyzed (~500). Although the data is very useful for quantifying the population fractions of lysed and PI-stained cells, the number of cells may not be enough to provide a statistically meaningful quantification for rare cells such as persisters and VBNC cells. The authors should mention this.*

We have now acknowledged that at high antibiotic concentrations (i.e. $> 6 \times \text{MIC}$) the fraction of persisters and VBNC cells drastically reduces (Johnson et al. PLoS Genetics 2013, 9, e1003123); therefore, we might not have been able to capture a statistically meaningful sample of persister and VBNC cells since we have analysed only 500 individual bacteria for each experimental condition.

This issue has been addressed on lines 378-382.

5. *Citations given in Lines 141-142: The studies of Dr. Wood (cited) focused on cell cultures that have been starved in PBS for weeks. The phenotypes they obtained are not VBNC cells. In fact, they are dead cells with empty cytosols. These cells can't be stained by PI because they don't have nucleotides. Unfortunately, their claims are not well justified.*

It has been identified many times by many groups that VBNC phenotypes with intact membranes and dense cytosols exist in normal cell cultures and they can be metabolically active. I would simply remove these citations from the text since their experimental conditions are not relevant to this study.

We have removed these citations.

6. *Higher concentrations of ampicillin and ofloxacin have reduced VBNC cell levels (almost zero, according to Figure 5). This is so interesting, any comment on this?*

We have now stated that it is interesting that at the highest antibiotic concentration employed (i.e. $25 \times \text{MIC}$) we did not detect any VBNC or persister cells. However, it is worth acknowledging (as also pointed out by this reviewer, please see point 4 above) that at high antibiotic concentrations (i.e. $> 6 \times \text{MIC}$) the fraction of persisters and VBNC cells drastically reduces (Johnson et al. PLoS Genetics 2013, 9, e1003123); therefore, we might not have been able to capture a statistically meaningful sample of persister and VBNC cells since we have analysed only 500 individual bacteria for each experimental condition.

This issue has been addressed on lines 377-382.

Reviewer 2.

1.1 The fact that all experiments were performed on stationary phase cells seems very important for all discussion and conclusions to be drawn from the results. It seems worth mentioning this even in the abstract and introduction and explaining the reasoning behind choosing stationary phase cells for this investigation.

We have now explained both in the abstract and the introduction that we studied stationary phase *E. coli*. We have also explained that we chose to investigate stationary phase *E. coli* because the fraction of phenotypic sub-populations (i.e. persister and viable but non culturable cells) that survive antibiotic treatments in this growth phase is typically larger compared to that measured for exponential phase *E. coli* (Goode et al. mBio 2021, 12, e00909-21; Keren et al. FEMS Microbiology Letters 2004, 230, 13).

This issue has been addressed on lines 17, 91-94, 111.

1.2 Related to this point, the abstract and results state that the capsids arrest growth within 2 hours (lines 24 and 113). Discussion of the ampicillin and ciprofloxacin phenotypes always refer to elongation rather than growth, and it seems that this may be a better word choice in the context of stationary phase cells.

We have now used the word elongation instead of growth throughout the manuscript as suggested by the reviewer.

1.3 It is not clear why growth, driven by new biosynthesis, should be occurring in stationary phase cells following antibiotic treatment, and elongation could alternatively be explained by turgor pressure acting on a weakened cell wall for example. It seems outside the scope of this paper to present evidence for one mechanism or another to explain the elongation phenotype, but it is worth explaining very clearly the context. These are all cells that have entered stationary phase and likely depleted an essential nutrient, and size increases could be explained by ongoing low levels of new biosynthesis, by changes in the cell wall allowing turgor-driven expansion, by changes in osmotic balance, or by some combination of these factors.

We have now added a sentence in the beginning of the results section to clarify that bacteria were simultaneously supplied with the antibiotic in use and lysogeny broth (LB) medium that permitted bacterial elongation and promoted exit from stationary phase.

This issue has been addressed on lines 116-117.

2. I find the distinction between VBNC and persister cells somewhat problematic. I know that this is following on from previously published work which used a similar designation. However, even that previous work also showed that "VBNC" and "persister" cells share many characteristics. The designation as "VBNC" or "persister" here is made based on whether a cell with an intact membrane after antibiotic exposure has started growing and dividing again or not at one time point (21 h) after removal of antibiotics. Others have shown that resumption of growth following antibiotic exposure can have a very long lag time (longer than 21 h), and that the distribution of lag times is not bimodal (for example, doi: 10.1038/s41586-021-04114-w) . There is no clear reason why the 21 h time point should be chosen as a threshold to distinguish between "persisters" (have resumed growth) and "VBNCs" (have not yet resumed growth). It seems possible that some of the VBNCs could instead be called persisters if the chosen time point for assessing resumption of growth was a bit later. My main point is that lag time before resumption of growth after surviving antibiotic exposure has been shown to have a broad, continuous distribution, but using a single time point to assess resumption of growth and assign cells to one of two categories suggests instead a binary outcome. It is interesting and relevant to evaluate the

impact of antibiotic type and concentration on survival and resumption of growth, but I think the use of the "persister" and "VBNC" categories is an oversimplification. To address this, I would suggest one of two options: 1) check whether the proportions of "VBNCs" and "persisters" remain unchanged at longer timepoints after antibiotic removal. If these designations remain stable for 36 or 48 hours after antibiotic removal, then this is outside the lag times previously reported and it is more convincing that the "VBNCs" really will not re-grow. 2) Add text to address the difficulty of designating any cell a "VBNC" with any certainty and acknowledge that any cell that still has an intact membrane but has not divided after 21 hours might be an example of a persister with a very long lag time, consistent with times that have been reported by others.

We have now added a sentence in the results section to clarify that we cannot exclude that the bacteria that we classified as VBNC cells were instead persisters with a very long lag time period (i.e. longer than 21h, our period of observation). In fact, lag times larger than 21h were recently reported for a minority of *E. coli* cells after starvation (Kaplan et al. Nature 2021, 600, 290).

This issue has been addressed on lines 149-151.

- 3. The asterisks used to denote significance do not seem to be explained anywhere (apologies if I have missed this). Which p-values do each number of asterisks refer to? I am not sure what the asterisks mean in the context of r values (lines 258-259 for example).*

We have now replaced asterisks with p-values throughout the manuscript, as also suggested by reviewer 1. We have also added to the results section the statistical test employed for statistical comparisons between distributions (i.e. unpaired, two-tailed, Welch's t-test) and for correlations between quantities (i.e. Pearson correlation coefficient r with two-tailed p-value).

December 19, 2022

Dr. Stefano Pagliara
University of Exeter
Living Systems Institute
Stocker road
Exeter EX4 4QD
United Kingdom

Re: Spectrum03667-22R1 (Single cell killing kinetics differentiate phenotypic bacterial responses to different antibacterial classes)

Dear Dr. Stefano Pagliara:

Congratulation. I am happy to inform you that your manuscript has been accepted, and I am forwarding it to the ASM Journals Department for publication. You will be notified when your proofs are ready to be viewed.

Sincerely,

Minsu Kim
Editor, Microbiology Spectrum
